Estimating bite force in extinct dinosaurs using phylogenetically predicted physiological cross-sectional areas of jaw adductor muscles

http://orcid.org/0000-0001-6447-406X Sakamoto Manabu msakamoto@lincoln.ac.uk
Department of Life Sciences, University of Lincoln , Lincoln , United Kingdom
De Baets Kenneth
Electronic publication date: 2022 Jul 12
Publication date: 2022
Volume: 10
Electronic Location ID: e13731
Received 2022 Apr 27; Accepted 2022 Jun 23
Copyright: © 2022 Sakamoto
Copyright year: 2022
Copyright holder: Sakamoto
License: This is an open access article distributed under the terms of the Creative Commons Attribution License, which permits unrestricted use, distribution, reproduction and adaptation in any medium and for any purpose provided that it is properly attributed. For attribution, the original author(s), title, publication source (PeerJ) and either DOI or URL of the article must be cited.
License URL: https://creativecommons.org/licenses/by/4.0/

Keywords: Bite force, Dinosaurs, Phylogenetic comparative methods, Phylogenetic predictive modelling, Physiological cross-sectional area, Biomechanics

Funding: The author received no funding for this work.

==============================
I present a Bayesian phylogenetic predictive modelling (PPM) framework that allows the prediction of muscle parameters (physiological cross-sectional area, APhys) in extinct archosaurs from skull width (WSk) and phylogeny. This approach is robust to phylogenetic uncertainty and highly versatile given its ability to base predictions on simple, readily available predictor variables. The PPM presented here has high prediction accuracy (up to 95%), with downstream biomechanical modelling yielding bite force estimates that are in line with previous estimates based on muscle parameters from reconstructed muscles. This approach does not replace muscle reconstructions but one that provides a powerful means to predict APhys from skull geometry and phylogeny to the same level of accuracy as that measured from reconstructed muscles in species for which soft tissue data are unavailable or difficult to obtain.

Introduction

Biomechanical modelling is an important means to infer the functional performances, ecologies, and behaviours of extinct animals for which such features cannot be directly observed (Anderson et al., 2012), e.g., in dinosaurs (Rayfield et al., 2001; Lautenschlager, 2013; Sakamoto, 2010; Gignac & Erickson, 2017; Bates & Falkingham, 2012). Biomechanical modelling can be particularly informative in terms of adaptive evolution and patterns of natural selection, when it outputs a univariate performance measure, such as bite force (Sakamoto, Ruta & Venditti, 2019). This is because performance measures like bite force represent tangible physical interactions with the environment in which the animals live or lived in. Bite force has repeatedly been reported as being correlated with dietary ecology in extant species (Dumont et al., 2012; Santana, Dumont & Davis, 2010; Santana, Strait & Dumont, 2011; Herrel et al., 2005a; Herrel et al., 2009), and thus has been treated as being likely informative for extinct species.

As bite force is the output of a musculo-skeletal lever system (Sinclair & Alexander, 1987), its estimation relies on input parameters including skeletal morphology and those derived from muscle anatomy and architecture, the latter of which is seldom preserved in fossils. Muscles thus need to be reconstructed first, then relevant muscle parameters estimated (Rayfield et al., 2001; Gignac & Erickson, 2017; Bates & Falkingham, 2012; Mazzetta, Cisilino & Blanco, 2004; Mazzetta et al., 2009). These parameters include the positions and orientations of muscle bodies, the weight, volume, and density of the muscle bodies, lengths of the muscle fibres, and the pennation angles of muscle fibres, all of which contribute to the physiological cross-sectional area (APhys) of muscles and capacity to generate force. However, muscle parameters based on reconstructions are associated with some degree of uncertainty (Gignac & Erickson, 2017; Bates & Falkingham, 2018). This owes to a number of reasons but chief among them is the unknowability of fiber lengths and pennation angles in fossil species. These parameters vary substantially amongst living species and are generally poorly documented. Fibre lengths and pennation angles (but especially the former) are crucial architectural data in estimating APhys, which itself being the determining factor of bite force. This is because force is proportional to APhys (Sinclair & Alexander, 1987) and thus the latter can be used to estimate isometric muscle force using a known stress factor σ (commonly 0.3 N/mm2).

Owing to difficulties and challenges facing muscle parameter reconstructions combined with the impact it has over downstream biomechanical modelling, there is need for a simple but reasonably accurate method of predicting APhys from skull geometries. Here, I present a Bayesian predictive modelling framework, the phylogenetic predictive model (PPM) (Organ et al., 2007; Sakamoto, 2021), to generate posterior predictive distributions of APhys from relationships between APhys and a skull geometry predictor variable, the skull width (WSk). Crucially, the aim of this article is not to present a method that accurately predicts APhys in fossil species from skull geometries as a substitute of muscle reconstruction, but a method that can predict APhys from skull geometries and phylogeny to the same level of accuracy as that measured from reconstructed muscles. Thus, the main objective is to provide the community with a tool to aid in reasonably accurate estimates of APhys in fossil organisms for which muscles are difficult to reconstruct.

Materials and Methods

Functional muscle groups

Jaw adductor muscles in archosaurs are largely grouped and named based on developmental biology and various topological criteria such as their relative positioning to nerves and blood vessels (Holliday & Witmer, 2007). However, from a functional perspective, the existing groupings are not necessarily congruent with lines of actions in a lever model. For instance, the Musculus (M) pseudotemporalis superficialis (mPSTs) is topographically and functionally similar to the M. adductor mandibularis externus (mAME) but are developmentally linked to the M. pseudotemporalis profundus (mPSTp), the latter of which is often physically connected to (and indistinguishable from) the M. adductor posterior (mAMP). This largely stems from the fact that the mPSTs and mAME both have cranial attachments in the temporal fossa, while the mPSTp and mAMP both attach onto the quadrate (Fig. 1). Thus, the mPSTs and mAME work together as inter-linked functional in-levers while the mPSTp and the mAMP work together as a separate set of inter-linked functional in-levers.

Figure 1 Jaw adductor muscles and functional muscle groupings in extant archosaurs.

(A) Attachment sites for jaw adductor muscles are depicted on a skull of a herring gull (Larus fuscus). Abbreviations are as follows: mAME, M. adductor mandibulae externus; mPSTs, M. pseudotemporalis superficialis; mPSTp, M. pseudotemporalis profundus; mAMP, M. adductor mandibulae posterior; and mPT, M. pterygoideus. Adductor muscle anatomy is then depicted for: (B) all adductor muscles; (C) temporal muscle group (mAME + mPSTs); (D) the quadrate muscle group (mPSTp + mAMP); and (E) the pterygoid muscle group (mPT).

I distinguish three functional adductor groupings, largely following Rayfield et al. (2001) and Sakamoto (2010), and identified muscle body complexes as follows: (1) the temporal muscle group (mTemp), consisting of mAME and mPSTs; (2) the quadrate muscle group (mQuad), consisting of the mPSTp and mAMP; and (3) the pterygoid muscle group (mPt), consisting of the M. pterygoideus (mPT). Practically, these approximate groupings are necessary as adductor muscles in smaller specimens are often difficult, if not impossible, to separate into the classic topological groupings, and as the goal of this study is to predict APhys in fossil species where we do not necessarily have detailed topological information. Furthermore, in the context of both biomechanical modelling and predictive modelling, approximation is often important in obtaining accurate predictions, which is not necessarily the most precise model.

Physiological cross-sectional areas in extant species

APhys for extant species (N = 39) were calculated from muscle architecture data collected predominantly from the literature but also from dissections (Sakamoto, Ruta & Venditti, 2019) (Struthio camelus, one specimen; Buteo buteo, three specimens; Larus fuscus, two specimens; Branta canadensis, one specimen; Gallus domesticus, two specimens; File S1; specimens were collected by the Bristol Ornithological Club and were donated to the University of Bristol as part of a clinical veterinary anatomy lab practical, c. 2005–2006 (Sakamoto, 2008)). APhys were calculated as: (1) APhys=(Mcosθ)/(ρL)

following Sacks & Roy (1982), where M is the wet weight of the muscle body (g), θ is the mean pennation angle, ρ is the specific density (1.056 × 10−3 g/mm3 (Burkholder et al., 1994)), and L (mm) is mean fiber length. In the case of parallel fibers θ is 0° and thus cosθ is 1.

Muscle measurements for APhys calculations were taken for two specimens of Buteo buteo and one specimen each of Larus fuscus and Struthio camelus. Muscles were weighed prior to sectioning to obtain M. Muscles were carefully sectioned under a microscope using a sharp scalpel. Incisions were made parallel to the length of the muscle fibers as much as possible. L and θ were measured using ImageJ (Rasband, 2012). As sectioning muscles can artificially truncate fibers and ends of fibers can often be obscured and difficult to determine, fiber lengths were taken at multiple locations on one or more sections through each muscle, the mean of which was taken as L.

In some specimens, APhys were approximated using the gross cross-sectional area (AGross), as simply the cross-section taken perpendicular to the long axis of the muscle body (Rayfield et al., 2001). AGross was measured in one specimen each of S. camelus, B. buteo, and Branta canadensis, and two specimens each of L. fuscus, and Gallus gallus. The muscle body was sectioned roughly perpendicular to the major axis of the muscle body at its widest point, and its AGross was digitally measured using Image J. The mean value of the left and right sides was taken as the final AGross value. Comparisons between AGross measurements and APhys calculations (following Eq. (1)) taken from different specimens within the same species, reveal that measured AGross values are generally congruent with calculated APhys values (Sakamoto, 2008; Persons & Currie, 2011; Snively & Russell, 2007).

Physiological cross-sectional areas in extinct species

For extinct archosaurs, cross-sectional areas of the jaw adductor muscles were estimated as AGross using a variant of the dry skull method (Thomason, 1991), whereby cranio-mandibular dimensions (namely the areas of the supratemporal, subtemporal, and mandibular fenestrae) were used to bound the AGross of individual jaw adductor muscles. I measured AGross of the mAME, mPSTs, mPSTp + mAMP and mPT on photographs and diagrams of reconstructed skulls taken at various angles of view. These are conceptually similar to previously published methods to estimate AGross in extinct dinosaurs (Rayfield et al., 2001; Mazzetta et al., 2009; Reichel, 2010). I further applied muscle pennation angle θ = 45° for the mTemp group, θ = 0° for the mQuad group, and θ = 30° for the mPt group, based on average pennation angles in my extant archosaur samples. I applied the effects of pennation on to AGross through division of AGross by sinθ. This approximates APhys in fossil archosaurs.

Predictor variable

I used the width of the skull (WSk) as the predictor variable in the phylogenetic predictive models (PPMs). WSk was chosen here as it has previously been demonstrated to predict bite force (Herrel et al., 2005a; Sakamoto, 2021; Herrel et al., 2005b) and various jaw adductor muscles well (Fig. 2). It is also readily available from the literature and easy to measure for a vast number of species for which muscle data do not exist, both extant and extinct. The utility of its wider applicability makes simple measures like WSk an ideal predictor in predictive modelling. WSk were mostly measured directly from osteological and fossil specimens where possible but augmented with data taken from photographs and literature.

Figure 2 Relationships between physiological cross-sectional areas and skull width in the PPM training set (N = 59).

Relationships between physiological cross-sectional areas APhys for each of the three muscle groups and skull width (WSk) are shown for extant (blue) and extinct (red) archosaurs in the PPM training set (N = 59): (A) temporal muscle group (mTemp); (B) quadrate muscle group (mQuad); and (C) pterygoid muscle group (mPt).

Phylogeny

I used an informal supertree of saurians based on the Time Tree of Life (TTOL) (Kumar et al., 2017) with fossil tips inserted manually at the appropriate phylogenetic locations (Sakamoto, Ruta & Venditti, 2019) (Fig. 3). Divergence times for fossil branches are based on first appearance dates (FAD) with terminal tips extended to their last appearance dates (LAD) using the paleotree R package (Bapst, 2012). I used the full range of temporal durations to scale the branches, as this allows for the maximum amount of time possible for trait evolution to occur (Sakamoto, Ruta & Venditti, 2019). Zero-length internal branch lengths were resolved by sharing time with neighbouring branches using the “equal” method (Bapst, 2012; Brusatte et al., 2008). While there are alternative methods to scaling branches, e.g., tip-dating using the fossilised birth-death model (Drummond & Stadler, 2016), phylogenetic regression under Brownian motion, which underlies the PPM framework, is extremely robust to uncertainties in branch lengths (Stone, 2011), so the choice of branch scaling makes minimal impact on PPM.

Figure 3 Phylogeny of extant and extinct saurians (N = 59) used in the phylogenetic predictive modelling.

The extant portion of the tree was taken from the TimeTree of Life and extinct tips inserted at the relevant positions.

Phylogenetic predictive modelling

I used a Bayesian PPM (Organ et al., 2007; Sakamoto, 2021) to predict APhys in extinct archosaurs. Separate PPMs were fitted on each of the three muscle groups as outlined above with the relevant APhys as the response variable and WSk as the predictor variable.

Model performance, or prediction accuracy, of each PPM was evaluated in a dataset containing only the extant species (N = 39) first, through Leave-One-Out Cross-Validation (LOOCV). LOOCV procedure largely follows that outlined in (Sakamoto, 2021), and is as follows: (1) the PPM was first fit on the dataset leaving one species out (N − 1) using Markov Chain Monte Carlo (MCMC) generating a posterior distribution of predictive models; (2) the posterior predictive models were used to predict APhys for the species that was left out from Step 1 based on the WSk and phylogenetic position of that species; (3) the posterior distribution of predictions (posterior predictive distribution) was evaluated against the actual APhys value recorded for that species. If the observed value fell outside the vast majority of the posterior predictive distribution (i.e., beyond 95% of the distribution; pMCMC < 0.05), then it is deemed that the actual APhys value is significantly different from the posterior predictive distribution, meaning that the prediction has failed in this particular species. I repeated these steps for all species in the data set (N = 39) and calculated the proportion of species for which the model succeeded in accurately predicting APhys out of the total sample size N.

I then predicted APhys for 53 fossil species of archosaurs (predominantly theropod dinosaurs). I first fitted a PPM on the N = 39 dataset and generated a posterior distribution of predictive models. I then used the predictive models to generate posterior predictive distributions for all 53 fossil species using their WSk and phylogenetic positions. This procedure is largely identical to LOOCV but is conducted in one step instead of one species at a time (Sakamoto, 2021). For 20 of the 53 fossil species, APhys measured from reconstructed muscles exist, thus allowing for assessment of the match between PPMs-predicted APhys and reconstructed APhys in fossil species.

Additionally, I evaluated prediction accuracy of PPMs on an expanded training set (N = 59) that includes APhys for select fossil species (N = 20) measured from reconstructed muscles (Sakamoto, Ruta & Venditti, 2019) or taken from literature (Lautenschlager, 2013; Gignac & Erickson, 2017; Bates & Falkingham, 2012; Lautenschlager et al., 2016). Prediction accuracy was evaluated through LOOCV as outlined above.

I then predicted APhys for the remaining 33 fossil species of dinosaurs (predominantly theropods). I first fitted a PPM on the N = 59 dataset and generated a posterior distribution of predictive models. I then used the predictive models to generate posterior predictive distributions of APhys for all 33 fossil species using their WSk and phylogenetic positions.

All model fitting was conducted in BayesTraits V3 over three independent MCMC chains each. The chains were run for 35,000,000 iterations, with the first 25,000,000 iterations discarded as burn-in, and sampled every 10,000 iterations after convergence, to produce a posterior sample of 1,000 predictive models and associated parameters.

Bite force estimation

Using the predicted APhys I estimated bite force (FBite) for 30 of the 33 fossil dinosaur species for which I predicted APhys through the PPM approach. I then compared those against FBite estimated for the 19 of the 20 fossil archosaurs based on measured APhys reconstructions included in the training set for the PPMs.

For each of the 30 fossil species for which I predicted APhys, I took the median value of the posterior predictive distribution for each muscle. Muscle force (FMusc) was then estimated for each muscle as the product of APhys and tetanic stress σ at 0.3 N/mm2 (or 300 kPa). The resulting FMusc was then multiplied by the muscle moment arm to yield the torque of that muscle. I measured relevant moment arms for each muscle following the procedures developed in Sakamoto (2010) (Fig. 4). Muscle moments were summed and divided by the distance between the fulcrum (jaw joint) and bite point and multiplied by two to derive a bilateral FBite. FBite was estimated for the anterior-most and posterior-most positions along the biting edge (tooth row or beak; Fig. 4).

Figure 4 Schematic depiction of a static lever model to estimate bite force in extinct dinosaurs.

Bite force (FBAnt and FBPost) was estimated in extinct dinosaurs using a static lever model as shown on a skull and mandible reconstruction of Deinonychus (author’s own work). FBAnt, anterior bite force; FBPost, posterior bite force; FTemp, temporal group muscle force; FQuad, quadrate group muscle force; FPt, pterygoid group muscle force.

FBite in fossil archosaurs that were included in the PPM training set (N = 19) were estimated in the same way as above but using APhys measured from reconstructed muscles as outline in Sakamoto, Ruta & Venditti (2019). I compared FBite at the posterior-most positions (maximum FBite) between the two sets of fossil species.

Results

Prediction accuracies of PPMs

Prediction accuracies of the PPMs in the initial training set consisting of only extant species (N = 39) was at 87% for all three muscle groups. The prediction accuracies of the PPMs in predicting APhys for the 20 fossil species, as compared to their measured APhys were 25%, 45% and 35%, respectively for the mTemp, mQuad and mPt groups.

Prediction accuracies of the PPMs in the expanded training set including fossil species (N = 59) were at 95%, 93% and 90% for the mTemp, mQuad and mPt groups, respectively. Out of the 20 fossil species included in the training set, in only two species (Plateosaurus engelhardti and Herrerasaurus ischigualastensis) did the PPMs fail to predict the observed APhys.

Bite force estimation

FBite estimated for the 30 fossil species based on predicted APhys are shown in Table 1. Compared to FBite estimated from reconstructed APhys in the 19 fossil species, these 30 FBite values fall along the expected relationship between FBite and WSk (Fig. 5). Comparisons between closely related species of similar sizes reveal the accuracy in resulting FBite values (Table 1): FBite for Deinonychus antirhoppus (predicted APhys) with WSk of 114.5 mm is 706N, while FBite for Dromaeosaurus albertensis (reconstructed APhys) with WSk of 103 mm is 885N; FBite for Carnotaurus sastrei (predicted APhys) with WSk of 300 mm is 7,172N while FBite for Majungasaurus crenatissimus (reconstructed APhys) with WSk of 300 mm is 7,845N.

Table 1 Bite forces estimated in extinct dinosaurs using APhys values either predicted through the PPMs or from reconstructed muscles.

Taxon	F BAnt	F BPost	W Sk	A Phys	
Acrocanthosaurus atokensis	8,266	16,984	480	Predicted	
Bambiraptor feinbergorum	50	97	55.5	Predicted	
Baryonyx walkeri	1,382	3,416	286	Predicted	
Carcharodontosaurus saharicus	11,312	25,449	558	Predicted	
Carnotaurus sastrei	3,392	7,172	300	Predicted	
Ceratosaurus nasicornis	2,432	5,998	270	Predicted	
Citipati osmolskae	202	225	77	Predicted	
Compsognathus longipes	8	15	24.6	Predicted	
Confuciusornis sanctus	12	17	31.3	Predicted	
Deinonychus antirrhopus	298	706	114.5	Predicted	
Dilong paradoxus	64	110	61.8	Predicted	
Eoraptor lunensis	35	95	40	Predicted	
Gallimimus bullatus	152	243	114	Predicted	
Garudimimus brevipes	121	183	84	Predicted	
Guanlong wucaii	268	512	124	Predicted	
Haplocheirus sollers	46	76	52	Predicted	
Incisivosaurus gauthieri	26	41	33.4	Predicted	
Monolophosaurus jiangi	1,710	3,872	243	Predicted	
Nanotyrannus lancensis	2,068	3,752	261	Predicted	
Nemegtomaia barsboldi	236	308	84	Predicted	
Ornithomimus edmontonicus	94	143	84	Predicted	
Shuvuuia deserti	12	15	31	Predicted	
Sinornithosaurus millenii	30	60	53.8	Predicted	
Spinosaurus aegyptiacus	4,829	11,936	451	Predicted	
Struthiomimus altus	108	187	80	Predicted	
Teratophoneus curriei	2,812	6,188	282	Predicted	
Tsaagan mangas	63	150	55	Predicted	
Velociraptor mongoliensis	131	304	91	Predicted	
Yangchuanosaurus shangyouensis	3,212	6,312	292	Predicted	
Zupaysaurus rougieri	325	1,012	119	Predicted	
Allosaurus fragilis	4,440	9,389	300	Reconstructed	
Archaeopteryx lithographica	2	3	16.8	Reconstructed	
Coelophysis bauri	72	289	74	Reconstructed	
Coelophysis rhodesiensis	99	393	77	Reconstructed	
Daspletosaurus torosus	8,385	16,641	525	Reconstructed	
Dromaeosaurus albertensis	443	885	103	Reconstructed	
Erlikosaurus andrewsi	118	229	100	Reconstructed	
Euparkeria capensis	86	216	49	Reconstructed	
Gorgosaurus libratus	6,418	13,817	467	Reconstructed	
Herrerasaurus ischigualastensis	678	1,937	107.7	Reconstructed	
Lesothosaurus diagnosticus	99	250	54	Reconstructed	
Majungasaurus crenatissimus	3,140	7,845	300	Reconstructed	
Ornithosuchus woodwardi	2,910	7,146	260	Reconstructed	
Parasuchus hislopi	450	1,958	183	Reconstructed	
Plateosaurus engelhardti	82	235	123	Reconstructed	
Riojasuchus tenuisceps	109	232	55	Reconstructed	
Sinraptor dongi	5,064	10,845	384	Reconstructed	
Tarbosaurus bataar	13,298	24,253	616	Reconstructed	
Tyrannosaurus rex	25,418	48,505	900	Reconstructed	
Note:

FBAnt, anterior bite force; FBPost, posterior bite force; WSk, skull width; and APhys, physiological cross-sectional area.

Figure 5 Relationship between bite force and skull width.

The relationship between bite force and skull width is shown for estimates based on predicted APhys (light green) and those based on muscle reconstructions (pink).

Discussion

The analyses presented here largely demonstrate two interesting features of predicting APhys in extinct species. First, the addition of APhys values for extinct species (measured from reconstructed muscles) in the training set drastically improved prediction accuracy of extinct species: compare 25–45% prediction accuracy using the extant-only PPM with 90% prediction accuracy (18/20 species) using the PPM that includes fossil species. The most likely cause of this improvement is increased sample size, from N = 39 to N = 59. It is well known and demonstrated through simulations that evolutionary parameters, such as Pagel’s λ (Pagel, 1997) lose statistical power with smaller sample size, with a marked reduction at approximately N < 50 (Freckleton, Harvey & Pagel, 2002; Sakamoto & Venditti, 2018). As the PPM approach taken here is also based on the Brownian motion model of phenotypic evolution, it has similar statistical properties to estimating λ, and would most likely encounter similar effects of sample size. Thus, increasing sample size to N = 59, which is just above this threshold of N = 50 previously suggested through both simulated (Freckleton, Harvey & Pagel, 2002) and empirical (Sakamoto & Venditti, 2018) cases, is likely the underlying cause for the improvement in prediction accuracy. Indeed, a similar recent study, which demonstrated that FBite estimated in extinct species are in line with those expected for animals of similar sizes, used a PPM based entirely on extant species but had a sample size of N = 188 (Sakamoto, 2021).

It is also possible however that the act of including fossil estimates in and of itself does have some positive effect on improving prediction accuracy. It has been shown before that inclusion of extinct tips in phylogenetic comparative analyses preserved phylogenetic signal λ (Pagel, 1997) in rates of phenotypic evolution deeper in the tree, while that of ultrametric trees degraded rapidly (Sakamoto & Venditti, 2018) – one interpretation is that subsequent evolution ‘overwrote’ signals from deeper in the tree when only data from extant taxa are modelled, but including fossil data deeper in the tree adds this information into the model. Thus, data associated with extinct tips that are deeper in time likely improves parameter estimation in phylogenetic comparative models. The oldest species in my dataset are approximately 250 million years old and comparatively close in time to the root of the tree and may contribute to this type of effect on evolutionary parameter (e.g., Brownian variance) estimation and the resulting posterior distribution of predictive models.

Crucially, as APhys in dinosaurs are generally much larger than those in most of the extant species in this dataset (Fig. 5), predicting for dinosaurs from PPMs based on the extant-only training set (N = 39) is effectively extrapolating far beyond the range of the data.

A note of caution however is that low prediction accuracy of APhys in extinct species using the extant-only training set may also be indicative of uncertainties related to muscle reconstructions based on skull geometries (Bates et al., 2021). However, high precision accuracy in the LOOCV of the N = 59 training set indicate that variation within APhys from muscle reconstructions in extinct species are within expected range of variance given phylogeny and Brownian motion (Sakamoto, 2021). Crucially, the objective of this study is to develop a method to predict APhys in extinct species from skull geometries that are within the same range of accuracy as those measured from reconstructed muscles, not to accurately predict real in-life APhys values as a substitute of muscle reconstruction – i.e., this method is to augment gaps in muscle reconstructed APhys data, not to replace muscle reconstructions entirely. Increased prediction accuracy by expanding the training set (N = 59) to include APhys estimates for 20 extinct species fulfils this purpose.

Second, the power of simple linear morphometrics (e.g., WSk) in predicting functionally important parameters is not to be taken lightly. The PPMs developed here are based only on WSk but is demonstrated to have prediction accuracy upwards of 95% depending on the muscle group. The fact that WSk is tightly correlated with FBite across multiple groups of vertebrates (Sakamoto, 2021; Herrel et al., 2005b) is consistent with these results. WSk is also tightly linked with body size, often scaling isometrically, making it the ideal predictor in PPMs for its ability to ground the model to a theoretical scaling framework, e.g., expected scaling exponent between area and length (Fig. 2). Simple metrics are also readily available across a wide taxonomic sample and can be collected from literature and osteological specimens, including fossils. PPMs based on such simple predictors are thus more versatile and robust.

Bite force estimates in extinct archosaurs

Using the APhys predicted from the PPMs, I estimated FBite in several extinct archosaurs. These values can be regarded as reasonably reliable estimates of true FBite in these extinct animals, given scaling relation with WSk and phylogeny. This owes to the fact that they are highly congruent with FBite estimates based on APhys measured from reconstructed muscles, which themselves have been demonstrated to be reasonably reliable estimates of FBite given size and phylogeny (Sakamoto, 2021). There are still some notable discrepancies between FBite estimated here with published values, namely in oviraptorosaurs (Meade & Ma, 2022) and ornithomimosaurs (Cuff & Rayfield, 2015), but also in Deinonychus (Gignac et al., 2010). My estimated FBite for oviraptorosaurs are under-estimates of published figures (Citipati osmolskae, 202–225N vs 349.3–499.0N (Meade & Ma, 2022); Incisivosaurus gauthieri, 26–41N vs 53–82.5N (Meade & Ma, 2022)) while those for ornithomimosaurs are over-estimates (Garudimimus, 121N at the tip vs 19N (Cuff & Rayfield, 2015); Ornithomimus, 121N vs 22N (Cuff & Rayfield, 2015); Struthiomimus, 108N vs 57.6N (Cuff & Rayfield, 2015)). The under-estimation of oviraptorosaurs is within the same order of magnitude (×101) and thus trivial in a comparative context (Sakamoto, 2021). On the other hand, the over-estimation of ornithomimosaurs is one order of magnitude (×102 vs ×101). The likely source of this discrepancy lies in the fact that neither oviraptorosaurs nor ornithomimosaurs were included in my N = 59 training set, thus making the predicted APhys reflecting values that are more typical of closely related theropod dinosaurs (Fig. 4; Table 1) as well as regressing to the mean of the training set (as is typical for regression). Future work expanding on the taxonomic sampling of the training set will undoubtedly improve prediction accuracy in specific taxa, especially in clades with unique dietary adaptations such as oviraptorosaurs (Sakamoto, 2010; Meade & Ma, 2022) or ornithomimosaurs (Cuff & Rayfield, 2015).

The case of discrepancy in FBite estimates for Deinonychus between this study and that by Gignac et al. (2010) warrants some special attention. Gignac et al. (2010) predicted the forces necessary to puncture bone to the depth observed in Tenontosaurus bones (Gignac et al., 2010; Erickson et al., 1996) and estimated the maximum FBite at the posterior-most biting position for Deinonychus at 8,200N, which is an order of magnitude higher than that estimated here at 706N (Table 1). My estimated FBite of 706N is in line with those of a similarly sized dromaeosaur Dromaeosaurus at 885N (WSk = 103 mm), and a similarly sized basal saurischian Herrerasaurus at 1,937N (WSk = 108 mm) (Table 1), but also extant diapsids with similar skull widths, Paleosuchus trigonatus at 1,082N (WSk = 121 mm), and Alligator sinensis at 1,084N (WSk = 122 mm) (Sakamoto, Ruta & Venditti, 2019). Interestingly the same can be said when the comparison is extended to extant carnivores with similar skull widths, Ursus thibetanus at 871N (WSk = 111 mm), Neofelis nebulosa at 1,296N (WSk = 119 mm), N. diardi at 1,117N (WSk = 115 mm), and Sarcophilus harrisii at 682N (WSk = 112 mm) (Sakamoto, Ruta & Venditti, 2019). It is important to note that forces necessary to puncture substrate are not confined to muscle-driven biting and may very well be the product of more aggressive kinetic feeding behaviours involving the whole head and neck (D’Amore & Blumenschine, 2009; Snively et al., 2013; Torices et al., 2018). This is supported by the observation that this bite mark in question was matched to a premaxillary tooth (Gignac et al., 2010), meaning that a long-snouted animal would have had to be capable of generating FBite of 3,000–4,000N (Gignac et al., 2010) at the tip of its snout through muscle generated biting, which is not congruent with the available data in comparably-sized amniotes (Sakamoto, Ruta & Venditti, 2019). Given its congruence with a similarly sized dromaeosaur Dromaeosaurus as well as other similarly sized amniotes, there is strong evidence to suggest that my FBite estimate for Deinonychus is reasonably accurate for an animal of its size.

Crucially, a comparison of FBite based on predicted APhys (N = 30) and FBite based on reconstructed APhys (N = 19) across the extinct non-avian theropods (Fig. 5; Table 1) generally show substantial overlap without any signs of sytemic biases (Fig. 5). This demonstrates that FBite estimates based on predicted APhys are neither systemically over- or under-estimates compared to FBite based on reconstructed APhys. Thus, PPMs are a useful approach to expand on reliable FBite data based on simple metrics and phylogeny to augment those based on reconstructed muscles. Importantly, this approach allows estimation of APhys and FBite in taxa (both extinct and extant) where muscle reconstruction is not feasible or possible for any number of reasons.

Discussion of FBite for individual species of interest are then valid and worth considering. Of note is that the large-bodied carnivorous dinosaurs, Carcharodontosaurus saharicus and Acrocanthosaurus atokensis, both reaching the size range of Tyrannosaurus rex, have FBite that are substantially lower than the latter, at 16,984 and 25,449N respectively, compared to 48,505N of T. rex. Carcharodontosaurus is approximately the same size as T. rex but is here shown to have had FBite that is approximately half of the latter. Carcharodontosaurus is typical in build and skull proportion for a theropod dinosaur, so the fact that its FBite was only half of that of T. rex is more likely a reflection of just how unique T. rex may have been compared to other theropods of similar sizes. Tyrannosaurus had robust conical-shaped teeth and multiple adaptations in the skull that allowed it to withstand immense forces (Cost et al., 2019; Rayfield, 2005; Snively, Henderson & Phillips, 2006). Multiple lines of evidence also point to habitual bone-crushing and -consumption in T. rex (Gignac & Erickson, 2017; Erickson et al., 1996; Erickson & Olson, 1996). These support the hypothesis that T. rex had at least a partial osteophagous diet, an ecology that was likely different from other theropods.

A similarly, large-bodied carnivorous dinosaur, Spinosaurus aegyptiacus, is here tentatively predicted to have had FBite at just under 12,000N, roughly in the same range as Sinraptor (10,845N), Gorgosaurus (13,817N), and Daspletosaurus (16,641N), all substantially smaller theropods. With the caveat that WSk for Spinosaurus was simply scaled up from the skull-width ratio of Suchomimus (Sereno et al., 1998), I offer additional support for this taxon to have had unique feeding habits for a theropod of its size. Spinosaurus shows adaptations in the craniomandibular morpho-functional complexes that are advantageous for generating relatively faster shutting speeds with less muscle input force (higher displacement advantage) at the expense of FBite (lower mechanical advantage) (Sakamoto, 2010). This would be congruent with a feeding mode relying on fast-snapping jaws rather than slow crushing bites, which is commonly observed in species with semi-aquatic feeding habits, including herons and egrets (Hone & Holtz, 2021).

Wider implications

The phylogenetic predictive framework I present here enhances collection of trait data that may be difficult to obtain across a wide taxonomic sample. Various in vivo measurements (e.g., bite force, muscle architecture) are obviously impossible to collect in extinct taxa, but they may also be difficult to obtain in many extant taxa, especially for those that are exotic, enigmatic, endangered, or dangerous. PPM then allows for predictions of trait data in such taxa provided that more readily available predictor variables (e.g., external physical traits) can be measured for them.

While the case study presented here focused on predicting muscle parameters, the PPM approach is not restricted to soft tissue predictions. The response variable Y of interest can be any single continuously varying trait as long as it exhibits significant relationships with a set of predictor variables Xi in taxa for which both Y and Xi can be measured. Examples include (but not precluded to): predicting body mass from skeletal measurements; predicting skeletal structural strength index from skeletal variables; or predicting metabolism from body mass.

Similarly, this framework is not restricted to any taxonomic group or scope. The taxonomic group of interest can be of any nature and breadth as long as a phylogeny exists that includes the taxa for which the Y variable is to be predicted.

A key feature of phylogenetic regression that makes the PPM framework extremely versatile is that it is extremely robust to uncertainties in terms of phylogenetic relations and branch lengths (Stone, 2011). This means that even a phylogeny with high levels of uncertainties (especially branch lengths for fossil trees) can be used effectively in a PPM framework to predict Y variables of interest. Nevertheless, modern approaches to branch scaling such as the tip-dating approach (Drummond & Stadler, 2016) that dates the divergences and tips simultaneously with topology inference (Gavryushkina et al., 2017; Stadler et al., 2018) can return a set of branch lengths that are guided by data and likelihood (i.e., probabilistic estimates), but these should be conducted using phylogenetic/cladistic data matrices (e.g., molecular or morphological characters), appropriate model of evolution, and tree prior, rather than on a fixed topology with no character data as is increasingly being used (Godoy et al., 2021; Gearty & Payne, 2020). The parameters associated with the latter are likely returning the priors as there are no data informing the likelihood, making the set of scaled branches no better than sampling randomly, but more investigation is needed.

It is important to note however that when considering Y and taxonomic sampling, the Y variable of interest should be broadly homologous across the phylogeny used, especially if the phylogenetic coverage is broad, spanning several higher order taxonomic groups, e.g., synapsids and diapsids. For instance, jaw adductor muscles of mammals (e.g., temporal and masseter muscles) are not directly comparable to those of diapsids (e.g., the mTemp, mQuad and mPt groups used here), but they are homologous within Mammalia across mammalian orders. On the other hand, biomechanical performance measures such as bite force is homologous across a very large portion of the tree of life, e.g., across vertebrates. The consideration here then would be the homology between the predictor variables across vertebrate clades with vastly diverse skull anatomy – e.g., skull measurements across various fish clades may not be precisely homologous, which is exemplified when the comparisons extend to amniotes.

There is no multi-response (multivariate regression) implementation as of date, but this is true for phylogenetic regression in general. This means that applications such as predicting morphospace coordinates would need to be conducted on individual shape (e.g., principal components) axis separately.

Conclusions

Here, I present a phylogenetic predictive modelling framework to predict soft tissue parameters (APhys) in extinct species from an osteological predictor variable (WSk). Predicted parameters are reasonably accurate given the known scaling relationship between the muscle parameter and predictor variable and phylogeny. Downstream biomechanical modelling yields performance metrics (FBite) that are in line with previous estimates based on muscle parameters from reconstructed muscles. Thus, phylogenetic predictive modelling provides a powerful means to predict soft tissue parameters for biomechanical modelling in extinct species from simple osteological predictor variables.

Supplemental Information

Supplemental Information 1 Raw data – APhys in extant reptiles.

Click here for additional data file.

Supplemental Information 2 Raw data – APhys in extinct archosaurs.

Click here for additional data file.

Supplemental Information 3 APhys data.

Click here for additional data file.

Supplemental Information 4 Phylogenetic tree.

Click here for additional data file.

Supplemental Information 5 LOOCV results N39 model.

Click here for additional data file.

Supplemental Information 6 LOOCV results N59 model.

Click here for additional data file.

Supplemental Information 7 Predicted APhys in fossil taxa.

Click here for additional data file.

Supplemental Information 8 Estimated bite forces for extinct taxa.

Bite forces estimated using either APhys predicted from the phylogenetic predictive models or from reconstructed muscles

Click here for additional data file.

Supplemental Information 9 Estimated bite force in theropod dinosaurs.

Click here for additional data file.

I would like to thank Chris Venditti for advice on phylogenetic comparative methods and building the phylogenetic predictive models and Andrew Meade for various technical support related to BayesTraits. I also am grateful for the positive and constructive comments made by Stephan Lautenschlager, Eric Snively and an anonymous reviewer that undoubtedly improved my manuscript. Finally, I’d like to thank the editor, Kenneth De Baets, for coordinating the review process.

Additional Information and Declarations

Competing Interests

Author Contributions

Data Availability

The author declares that he has no competing interests.

Manabu Sakamoto conceived and designed the experiments, performed the experiments, analyzed the data, prepared figures and/or tables, authored or reviewed drafts of the article, and approved the final draft.

The following information was supplied regarding data availability:

The data is available at Open Science Framework: Sakamoto, Manabu. 2022. “Estimating Bite Force in Extinct Dinosaurs Using Phylogenetically Predicted Physiological Cross-Sectional Areas of Jaw Adductor Muscles.” OSF. April 23. DOI 10.17605/OSF.IO/VXD59.

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
