# Peer review of "Estimating bite force in extinct dinosaurs using phylogenetically predicted physiological cross-sectional areas of jaw adductor muscles"

_PeerJ, doi:10.7717/peerj.13731_

## Round 0.1 · original submission · Minor Revisions

You present a novel and very valuable approach to estimate bite forces in extinct species which can also be extended to modern species for which traditional bite force experiments are not feasible. The manuscript is very well written and a solid contribution to the field as demonstrated by the unanimous opinions by the reviewers. I would like to see this published, but there are some minor but to some degree crucial points I would like to see addressed before publication. These main points being:

Comparison between predicted and existing estimates: I agree with reviewer 1 that the manuscript would greatly benefit from such comparison to establish if the method under- or overestimates bite forces and what causes these differences (e.g., Deinonychus: compare reviewer 2). A more discussion concerning the accuracy of the method would also be valuable (compare reviewer 3).

Usefulness and applicable of the approach: the herein applied method allows to estimate jaw adducting muscle cross-sectional area based on simple linear measurement and in turn bite forces after corrections for phylogeny. In this way, performance estimates for some taxa enhance estimates for other extinct taxa. I feel these characteristics should be highlighted more (compare reviewer 2). In this context, it would also be valuable to have a broader discussion on how efficiently the model/approach which is applied here to dinosaurs could be transferred to mammals and if the approach would work or can be modified to work across a wide range of phylogenetically different species (compare reviewer 1)

Branch lengths: please elaborate on differences between branch lengths and fossil record and how later studies with refined branch lengths could improve the methods (compare reviewer 2)

Citations: some additional relevant papers could be cited (compare reviewer 2)

Please make sure these as well as all other points raised by the reviewers including those in annotated pdfs are addressed.

I look forward to receiving the revision.

·

Basic reporting

The manuscript is very detailed and clear. The text is comprehensive, provides a good background on the topic supported by relevant and adequate references and overall very professionally written.

Experimental design

The methodology is very interesting and offers and alternative and novel approach to estimated bite forces in extinct species and could also be applicable to modern species for which traditional bite force experiments are not possible.
The methods are described in detail and have been performed to a professional standard.

Validity of the findings

The results are clear and reported in detail. Their discussion is valid and supported by the results.

Additional comments

This is a very interesting study providing a new outlook to estimate bite forces in fossil species. I found the manuscript enjoyable to read and think this is a worthwhile contribution to the topic.

I have one more substantial point I would like to see included/discussed: How do the predicted bite force values obtained in this study compare to existing estimates? I appreciate that not all studied species have been the focus of previous work but there seem to be some differences in specific taxa. For example: https://www.nature.com/articles/s41598-022-06910-4 and https://peerj.com/articles/1093/
It seems the method under-/overestimates bite forces and a discussion where these differences could stem from would be useful.

Apart form that, I have a few minor points, questions, and suggestions but otherwise find the manuscript ready to be accepted.

Individual points:

- Line 40: Add here that bite force also depends on the skeletal morphology, not only on muscle anatomy and architecture alone.

- Line 114: Which part of the muscle body was sectioned? The region with the highest circumference?

- Line 116: Mention here what these calculations were based on.

- Line 126/128: "I" not "We"

- Line 132: Maybe explain the abbreviation PPM here again (I know it is in the introduction but the reader may have forgotten at this point).

- Line 281ff: Could you discuss briefly, how specific the model used here is. For example, is it only valid for dinosaurs or could it be transferred easily to modern and fossil mammals? Further, would it work of the sample size included a mix of species across a wide range of phylogenetically diverse species?

- LIne 314ff: Kind of following on from the previous comment. How adaptive is the model in general? Could other performance measures be analysed in a similar fashion? What would be required to do that?

·

Basic reporting

The basic reporting is excellent in all categories. I strongly suggest citing a few additional papers. Absolutely minimal suggestions for the writing.

Experimental design

The goal of the paper is so important that I suggest putting it up front in the abstract. The hourglass structure of scientific writing (broad contextualization to focus to broad implications) is fine, but we tend to bury the real lede. The reader might go from "bite force" to "Spinosaurus" and miss the paper's strongest value (see Validity below).

The commented manuscript calls attention some muscle vectors, which appear to extend farther anteriorly (if they were position vectors) than intuitively likely centroids of muscle pull (Cost et al. 2019, 2022).

Validity of the findings

The paper is tremendously useful in its findings that:
1. we can estimate bite forces with phylogenetic prediction based on an easy measurement (skull width);
2. sample size and performance estimates for some extinct taxa powerfully enhance estimates for other extinct taxa.

The predicted estimates in Table 1 are useful and interesting on their own, and the discussion of examples is appropriate and fascinating.

The dissections and measurements of muscle PSCA parameters are terrific as primary data and validation. I really like the groupings of muscle by in vivo function, rather than developmental classification.

Additional comments

The manuscript was interesting both operationally and philosophically. As suggested above, put the paper's rationale and goal in the first sentence of the abstract. I learned in engineering that if you bury the lede, people die. State the result or purpose immediately, and then (like good scientists) we can introduce the history of bridge building.

A peripheral and broad issue I'd like addressed is estimates of branch lengths from first and last occurrences. I can look at a familiar part of the tree in Figure 3 and see branch lengths that appear to radically contradict fossil evidence, and what we'd expect from finer-scaled phylogenies with more taxa.

I suggest commenting on this as a potential issue for the results, and cite it as a real strength of the method: later studies with refined branch lengths are likely to improve already excellent phylogenetic predictions. Or, tell me I'm wrong. Likely as not, but I've spent hours correcting occurrences from the Paleobiology Database.

If the author wishes, they might address the 5-10 fold discrepancy of bite force estimates for Deinonychus from the current method and bone indentation experiments. The issue is probably outside the scope of the paper.

The manuscript is an innovative, useful, and important methodological and biological contribution that I'd like to see published quickly.

Reviewer 3 ·

Basic reporting

This is a well conceived and generally well written MS and I have no significant concerns with the design or findings. The author presents a method whereby researchers can generate predictions of jaw adducting muscle cross-sectional area based on simple linear measurements after correcting for phylogeny. These in turn can be used as bases for predicting bite forces. Results are largely in line with those based on actual muscle reconstructions.
This certainly provides a useful tool. However, it is also clear that in some instances the proposed method is less accurate than those based on reconstructions. This is acknowledged in the MS, but I do think that this should be clearly stated from the get go, i.e., in the Abstract. It may well be a minority of species to which this applies, but ultimately estimates of cross-sectional area based on reconstructions will be more reliable.
Couple of very minor points:
Line 90-91: "APhys for extant species were calculated from muscle architecture data collected predominantly from literature but also from dissections..." The actual number of extant species and range of taxa should be given here. As it is it's only available in the Suppl Data. "the" should be inserted before "literature".
Line 126: "We further applied muscle pennation...." This is a single author MS.

Experimental design

no comment

Validity of the findings

no comment

---

## Round 0.2 · Minor Revisions

Thank for addressing all the suggestions. The paper has become even more comprehensive and easier to follow. You paper is as good as accepted. I only feel it would be appropriate to have at least one reference per group corroborating/discussing the "unique dietary adaptations such as oviraptorosaurs or ornithomimosaurs." Likely this is trivial to you but might not be to all readers. I look forward to receiving the revised manuscript and seeing it published.

---

## Round 0.3 · accepted · Accept

Thank you for addressing these final suggestions. I look forward to seeing this work published.